# Liver function tests and fibrosis scores in a rural population in Africa: a cross-sectional study to estimate the burden of disease and associated risk factors

Geraldine O'Hara,[1] Jolynne Mokaya,[2] Jeffrey P Hau [ORCID],[1,3] Louise O Downs,[2,4] Anna L McNaughton,[2] Alex Karabarinde,[3] Gershim Asiki,[3] Janet Seeley,[3,5] Philippa C Matthews [ORCID],[2,4,6] Robert Newton[3,7]

GO and JM contributed equally. PCM and RN contributed equally.

For numbered affiliations see end of article.

**Correspondence to**
Dr Philippa C Matthews;
philippa.matthews@ndm.ox.ac.uk

## ABSTRACT

**Objectives** Liver disease is a major cause of morbidity and mortality in sub-Saharan Africa, but its prevalence, distribution and aetiology have not been well characterised. We therefore set out to examine liver function tests (LFTs) and liver fibrosis scores in a rural African population.

**Design** We undertook a cross-sectional survey of LFTs. We classified abnormal LFTs based on reference ranges set in America and in Africa. We derived fibrosis scores (aspartate aminotransferase (AST) to Platelet Ratio Index (APRI), fibrosis-4, gamma-glutamyl transferase (GGT) to platelet ratio (GPR), red cell distribution width to platelet ratio and S-index). We collected information about alcohol intake, and infection with HIV, hepatitis B virus (HBV) and hepatitis C virus (HCV).

**Setting** We studied a population cohort in South-Western Uganda.

**Participants** Data were available for 8099 adults (median age 30 years; 56% female).

**Results** The prevalence of HBV, HCV and HIV infection was 3%, 0.2% and 8%, respectively. The prevalence of abnormal LFTs was higher based on the American reference range compared with the African reference range (eg, for AST 13% vs 3%, respectively). Elevated AST/ALT ratio was significantly associated with self-reported alcohol consumption (p<0.001), and the overall prevalence of AST/ALT ratio >2 was 11% (suggesting alcoholic hepatitis). The highest prevalence of fibrosis was predicted by the GPR score, with 24% of the population falling above the threshold for fibrosis. There was an association between the presence of HIV or HBV and raised GPR (p=0.005) and S-index (p<0.001). By multivariate analysis, elevated LFTs and fibrosis scores were most consistently associated with older age, male sex, being under-weight, HIV or HBV infection and alcohol consumption.

**Conclusions** Further work is required to determine normal reference ranges for LFTs in this setting, to evaluate the specificity and sensitivity of fibrosis scores and to determine the aetiology of liver disease.

## Strengths and limitations of this study

► This is a cross-sectional study of a large well defined population cohort in rural South-Western Uganda where the burden of liver disease and its aetiology is not well described.

► Our cross-sectional analysis of liver function tests (LFTs) and fibrosis scores provides insights into some of the risk factors for liver disease, allowing us to make preliminary estimates of the burden of liver disease, and particularly highlighting a significant contribution of alcohol.

► LFTs are a blunt tool for assessment of liver health, with many potential confounding factors. This current study only accounts for a limited range of aetiological agents.

► LFTs were measured at only one point in time, potentially overcalling liver disease as a result of transient abnormalities.

► A high HIV prevalence may be a confounding factor, causing abnormalities in platelet counts and elevation in LFTs that may not correlate well with underlying liver disease.

## INTRODUCTION

Liver disease causes an estimated 200 000 deaths each year in sub-Saharan Africa (sSA) as a result of liver cirrhosis and hepatocellular carcinoma.[1] More than 80% of Africa's burden of liver disease has been attributed to endemic blood borne virus (BBV) infections, such as HIV, hepatitis B (HBV) and hepatitis C (HCV), alcohol, hepatotoxic medications (including traditional and herbal medicines), non-alcoholic fatty liver disease (NAFLD) and exposure to aflatoxins.[1–3] However, the prevalence, distribution and aetiology of liver disease in many parts of Africa have not been well characterised, and the neglect of cirrhosis has recently been highlighted.[2] In order to improve screening for liver disease, and to implement appropriate investigations and intervention, we have undertaken a survey of liver function tests (LFTs) together

BMJ

with demographic data for a large rural cohort in South-Western Uganda.[4]

The term 'LFTs' can be ambiguous, as it is widely applied to biochemical markers of liver inflammation or biliary obstruction, rather than genuine hepatic function. These include aspartate aminotransferase (AST), alanine aminotransferase (ALT), alkaline phosphatase (ALP), gamma-glutamyl transferase (GGT) and bilirubin (BR). This panel of blood biomarkers is usually the first approach to the evaluation of liver disease; reference ranges and causes of derangement are summarised in online supplementary table 1.[5] In addition, true tests of liver synthetic function can be assessed by measuring prothrombin time or albumin, and platelet production may be decreased in chronic liver disease due to hypersplenism, decreased thrombopoietin levels and bone marrow suppression.[6] Abnormal LFTs are often non-specific and can arise transiently in association with many acute illnesses or usage of medications. However, persistent derangement can indicate chronic liver disease, with associated morbidity and mortality.[7] The pattern of derangement can sometimes help to establish aetiology—for example AST/ALT ratio >2 is characteristically associated with alcoholic hepatitis.[8 9]

Determination of the origin of liver disease and stratification for treatment necessitate estimation of the extent and nature of hepatic injury. Liver biopsy allows objective grading of fibrosis and can provide information about the likely aetiology of liver disease based on specific changes to cellular architecture. However, biopsy is costly, requires experts to undertake the procedure and analyse samples, and exposes patients to potentially life-threatening risks. Imaging can also be employed to assess fibrosis. Typically, this comprises ultrasound-based techniques, including fibroscan to derive elastography scores. In most low-income and middle-income settings, evaluation of liver disease currently depends on use of non-invasive (blood) markers, often combined with ultrasound and/or fibroscan when available.

Non-invasive fibrosis blood tests are simple and offer a safe route to assess for liver fibrosis, appealing in resource-limited settings. Scores of liver fibrosis, such as AST to Platelet Ratio Index (APRI), fibrosis-4 (FIB-4), GGT to platelet ratio (GPR), red cell distribution width to platelet ratio (RPR) and S-index have been derived using liver enzymes (ALT, AST, GGT) in combination with platelet count. However, diagnostic accuracy is not well established in sSA and can be influenced by the population being assessed and the nature of underlying liver disease.[10–15] GPR has recently been reported as an independent predictor of significant fibrosis in naive Gambian patients with chronic hepatitis B (CHB) infection,[13] while the usefulness of cut-off values for APRI scores in CHB has been questioned.[16] However, further studies are needed to determine the specificity and sensitivity of different scores in different settings.

Appropriate reference ranges for LFTs are crucial for optimising the detection of underlying liver disease.[17]

Application of reference ranges defined in one setting to different populations may lead to either underestimation or overestimation of abnormalities.[17–19] As well as being dependent on the population being assessed, the distribution of LFTs in any given setting can also be influenced by the type of instrument, reagents used and the strength of quality assurance.[19] Efforts have been made to establish 'population-specific' reference ranges[18 20]; one example is through the application of cross-sectional data from seven South-Eastern African countries.[18] However, such local reference ranges for Africa have been derived from cross-sectional data collected in adults without addressing the potential prevalence of underlying liver disease. Thus, while American reference ranges potentially overestimate of the burden of liver disease in an African setting, it is also possible that locally derived reference ranges underestimate the burden (as they are based on thresholds that have been derived from populations in which liver disease is highly prevalent).

We here set out to assess LFTs and fibrosis scores derived from a large, well defined population cohort in rural South-Western Uganda.[21] We applied reference ranges set in both America and in Africa,[18 22] in order to assess the possible burden of liver disease, highlighting the discrepancies that arise as a result of the difference between thresholds. We derived fibrosis scores to further evaluate the potential prevalence of liver disease in this setting and to estimate the contributions of alcohol and BBVs to the burden of disease.

## METHODS

### Study design and study population

We conducted a cross-sectional study in a rural population in Kyamulibwa, in the Kalungu district of South-Western Uganda as part of the survey of the General Population Cohort (GPC). The GPC is a community-based cohort established in 1989 with funding from the UK Medical Research Council (MRC) in collaboration with the Uganda Virus Research Institute (UVRI).[23] Regular census and medical surveys have been conducted in this population cohort. In 2011, data collection included screening for viral hepatitis and LFTs among 8145 adults (≥16 years), which we used for this analysis.

### Data collection

Demographic and health history data were collected using questionnaires and standardised procedures described elsewhere.[23 24] Blood samples were drawn at home and transported for testing at the MRC/UVRI and London School of Hygiene and Tropical Medicine (LSHTM) central laboratories in Entebbe. LFTs (serum AST, ALT, ALP, GGT and BR) were measured using a Cobas Integra 400 plus machine, with Roche reagents. Screening for HIV testing was done using an algorithm recommended by the Uganda Ministry of Health, based on initial screening with a rapid test. If the test result

was negative, the participant was considered to be HIV negative. If the test result was positive, the sample was retested with the rapid test HIV-1 or HIV-2 Stat-Pak. If both tests resulted in a positive result, the participant was diagnosed as HIV positive. If the tests gave discordant results, the sample was further evaluated with the rapid test Uni-Gold Recombinant HIV-1/2. For those samples assessed by all three tests, two positive test results were interpreted as positive, and two negative results were considered negative. HBV surface antigen (HBsAg) testing was conducted using Cobas HBsAg II (2011–08 V.10), and those who tested positive were invited for further serologic testing. HCV was tested using a combination of immunoassays followed by PCR, as previously described.[25] Normal serum levels of liver enzymes were classified according to the American reference range (ARR, MGH Clinical Laboratories) and local reference ranges (LRR[18]; values are listed in online supplementary table 1).[5] We excluded individuals ≤19 years from ALP analysis, since elevated ALP can be attributable to bone growth in teenagers.

Data from the full blood count was used to calculate fibrosis scores (mean corpuscular volume and platelet count). This was collected starting part-way through the 2011 data collection period; the data are, therefore, population-based, although based on only a subset of the whole cohort (n=1877).

## Calculation of fibrosis scores and AST/ALT ratio

Where data were available (n=1877), we calculated APRI, FIB-4, GPR, RPR and S-index. The formulae for calculating these scores are presented in online supplementary table 2,[5] along with the sensitivity and specificity of each, based on previous studies. We used previously established thresholds to suggest the presence of liver fibrosis, as follows: APRI>0.7,[26] FIB-4>3.25,[27] GPR>0.32,[13] RPR>0.825[28] and S-index>0.3.[29] We also calculated the AST/ALT ratio; a score >2 has been associated with alcoholic hepatitis.[9]

## Statistical analysis

We analysed data using standard statistical software, Stata/IC V.13 (Stata Corporation, College Station, USA) and GraphPad Prism V.7.0. We summarised participant baseline characteristics using proportions (%) and these were stratified by sex. We reported prevalence and distribution of LFTs, laboratory markers of fibrosis and elastography scores using descriptive statistics. We reported p values from $\chi^2$ tests, comparing the proportions of each potential risk factor between male and female participants. We also reported the medians and IQRs of each LFT and liver fibrosis scores. We compared the difference in medians of LFTs and liver fibrosis scores for each potential risk factor using the Kruskal-Wallis test.

We used logistic regression in our univariate and multivariate analyses, using the threshold for significance set at 0.05, to estimate the ORs, along with its 95% CIs, to identify potential factors associated with abnormal LFTs and

liver fibrosis scores, using a forward stepwise approach to develop our multivariate models. We added risk factors that were identified in the age-adjusted and sex-adjusted analysis to the multivariate model. The final multivariate models for each LFT and liver fibrosis score were obtained by excluding variables in the final model until all remaining variables were associated with abnormal LFTs and liver fibrosis scores at the p<0.05 threshold. Once the final multivariate model had been established, variables that were eliminated through this forward stepwise approach were added back to the model and were reported if associated at the p<0.05 threshold, to reduce the effects of residual confounding.

Due to the low number of individuals with active HCV infection at the time of the study, we did not include this subgroup in univariate or multivariate analysis. These HCV RNA-positive individuals have been described in more detail elsewhere.[30] We present results of multivariate analysis in the form of Forrest plots generated using Microsoft Excel v.16.

We calculated population attributable risk (PAR) as the proportion of the cases of liver dysfunction (defined either as elevated LFTs or fibrosis score) in the population that is due to exposure to alcohol, HIV or HBV. This provides us with an estimate of the proportion of liver dysfunction that would be eliminated if exposure was removed.[31]

## Patient and public involvement

Patients and the public were not involved in the design, conduct or reporting of this research.

## RESULTS

### Characteristics of study population

We analysed complete data for 8099 participants (summarised in online supplementary table 3).[5] Compared with females, there were more males who were HBV positive, (prevalence 3% vs 2%, respectively; p<0.001) and had consumed alcohol in the past 30 days, (40% vs 33%, respectively; p<0.001). More females were HIV positive (9% vs 6%, respectively; p<0.001). Males were more likely to be underweight (31% vs 16%) and females to be overweight (18% vs 5%); p<0.001 in both cases. Median and IQR for each parameter analysed are presented in online supplementary table 4.[5]

### Proportion of population defined as having abnormal LFTs varies according to the reference range that is applied

The proportion of the population falling above the upper limit of normal (ULN) for each parameter is shown in table 1, with ALT, AST and GGT distributions in figure 1A–C (full data for all LFTs are shown in online supplementary figure 1).[5] These results highlight the different burden of disease that can be estimated according to the reference range that is applied, with a higher proportion of the population falling above the ULN when the ARR was applied compared

**Table 1** Study participants from the Uganda GPC with abnormal LFT results and fibrosis scores based on ULN according to American Reference Range (ARR) and Local Reference Range (LRR)

| Enzyme type | Total n/N (%) | Male n/N (%) | Female n/N (%) | P value* |
|---|---|---|---|---|
| **ALT** | | | | |
| Abnormal ARR† | 573/8099 (7.1) | 162/3542 (4.6) | 411/4557 (9.0) | <0.001 |
| Abnormal LRR‡ | 209/8099 (2.6) | 87/3542 (2.5) | 122/4557 (2.7) | 0.53 |
| **AST** | | | | |
| Abnormal ARR† | 1011/8099 (12.5) | 434/3542 (12.3) | 577/4557 (12.7) | 0.58 |
| Abnormal LRR‡ | 241/8099 (3.0) | 123/3542 (3.5) | 118/4557 (2.6) | 0.02 |
| **GGT§** | | | | |
| Abnormal ARR† | 889/8099 (11.0) | 362/3542 (10.2) | 527/4557 (11.6) | 0.06 |
| **BR** | | | | |
| Abnormal ARR† | 1051/8099 (13.0) | 635/3542 (18.0) | 416/4557 (9.1) | <0.001 |
| Abnormal LRR‡ | 497/8099 (6.1) | 214/3542 (6.0) | 283/4557 (6.2) | 0.75 |
| **ALP¶** | | | | |
| Abnormal ARR† | 1161/5616 (20.7) | 315/2273 (13.9) | 846/3343 (25.3) | <0.001 |
| Abnormal LRR‡ | 139/5616 (2.5) | 60/2273 (2.6) | 79/2273 (2.4) | 0.513 |
| **FIB-4** | | | | |
| Abnormal** | 99/1877 (5.3) | 54/824 (6.6) | 45/1053 (4.3) | 0.03 |
| **APRI††** | | | | |
| Abnormal ARR*,** | 145/1877 (7.7) | 95/824 (11.5) | 50/1053 (4.8) | <0.001 |
| Abnormal LRR*,** | 60/1877 (3.2) | 42/824 (5.1) | 18/1053 (1.7) | <0.001 |
| **GPR** | | | | |
| Abnormal** | 441/1877 (23.5) | 185/824 (22.5) | 256/1053 (24.3) | 0.35 |
| **AST/ALT** | | | | |
| Abnormal** | 882/8099 (10.9) | 420/3542 (11.9) | 462/4557 (10.1) | 0.01 |
| **S-index** | | | | |
| Abnormal** | 73/1877 (3.9) | 50/824 (6.1) | 23/1053 (2.2) | <0.001 |

*P value calculated to determine whether significant difference was observed between males and females in each category using $\chi^2$ test.
†Abnormal LFTs, according to American Reference Range, ARR, are defined as test results outside of the following ranges: ALT (male: 10–55 U/L, female: 7–30 U/L), AST (male: 10–40 U/L, female: 9–32 U/L), GGT (male: 8–61 U/L, female: 5–36 U/L), BR (0–17 mmol/L) and ALP (male: 45–115 U/L, female: 30–100 U/L).
‡Abnormal LFTs, according to Local Reference Range, LRR, are defined as test results outside of the following ranges: ALT (8–61 U/L), AST (14–60 U/L), BR (2.9–37 mmol/L) and ALP (48–164 U/L).
§ LRR for GGT not defined.
¶Individuals under the age of 19 were excluded.
**Threshold usedto predict liver fibrosis: APRI > 0.7; FIB-4 >3.25; GPR >0.32; RPR >0.825; S-Index >0.3.
††APRI score calculated using ULN of AST using both the ARR and LRR.
ALP, alkaline phosphatase; ALT, alanine transaminase; APRI, AST to Platelet Ratio Index; ARR, American reference range; AST, aspartate transaminase; AST/ALT ratio, aspartate/alanine ratio; BR, total bilirubin; FIB-4, fibrosis 4; GGT, gamma-glutamyl transpeptidase; GPC, General Population Cohort; GPR, GGT to platelet ratio; LFT, liver function test; LRR, local reference ranges; RPR, red cell distribution width to platelet ratio; ULN, upper limit of normal.

with the LRR (figure 1A, B). Most striking, for AST, 13% of the population had a value that was deemed to be elevated based on ARR, compared with only 3% based on the LRR (figure 1B). Using the ARR, ALT and BR were significantly more likely to be above the ULN in males than in females, and ALP was more likely to be higher in females (p<0.001 in each case, table 1). These sex differences were not apparent when the LRR was applied. OR for deranged LFTs and fibrosis scores according to age and sex is shown in online supplementary figure 2.[5]

### The highest prevalence of liver fibrosis is predicted using the GPR score

We calculated APRI, FIB-4, GPR, RPR and S-index scores (table 1). The estimated prevalence of fibrosis was highest when based on GPR score (23.5%; figure 1D), compared with FIB-4 (5.3%), APRI (3.2%), S-index (3.9%) and RPR (0.1%). We excluded RPR scores from further statistical analysis because only few individuals were classified as having an elevated score (we therefore did not have statistical power to detect any factors associated with abnormal score). Because the APRI is derived using the

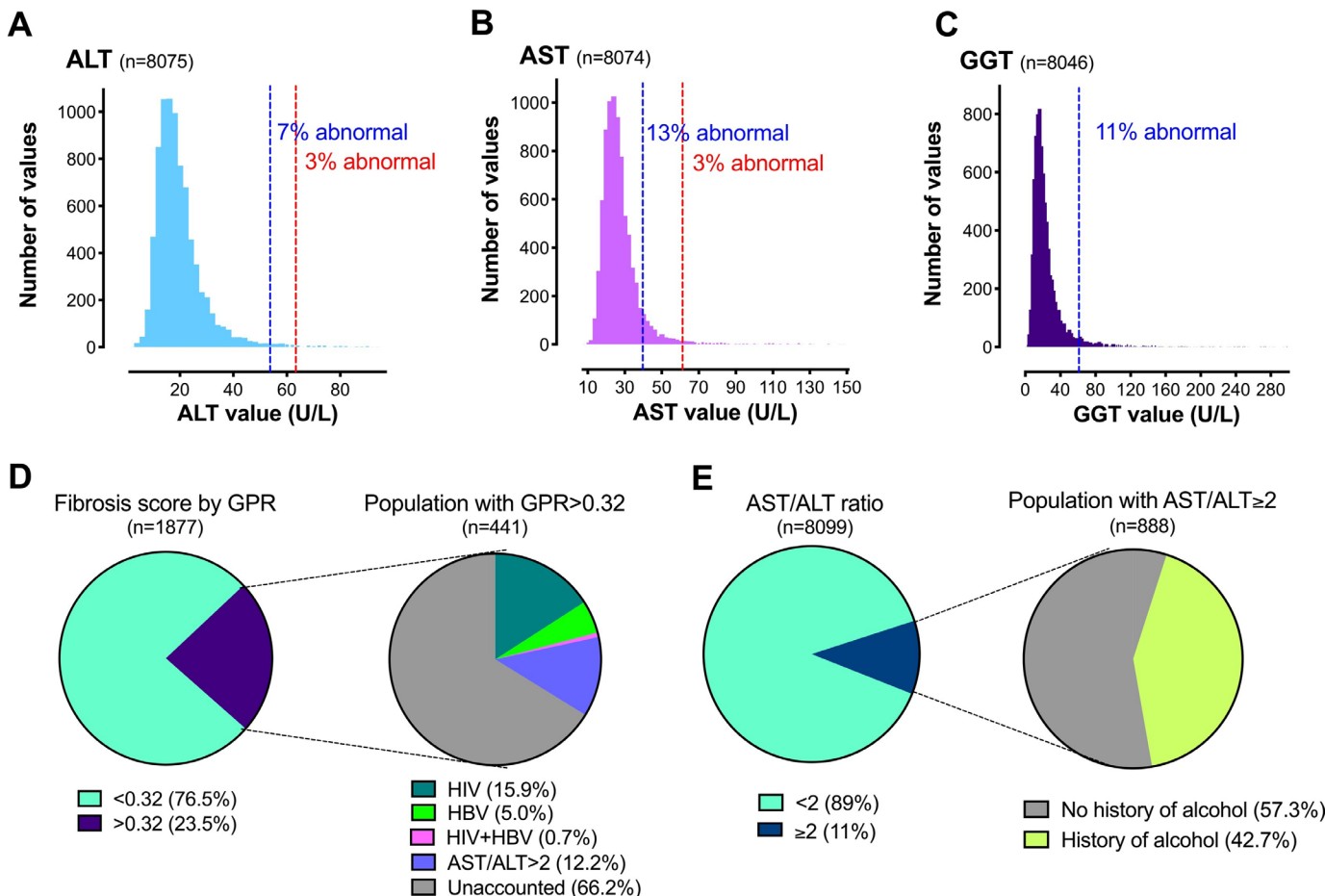

**Figure 1** LFTs and hepatic fibrosis scores among adults in the Uganda GPC. Distribution of (A) ALT, (B) AST and (C) GGT. Dashed vertical lines indicate ULN based on American Reference Range, ARR (blue) and Local Reference Range, LRR (red), as shown in online supplementary table 2.[5] Note no LRR defined for GGT. (D) Proportion of the population with an elevated GPR score, and among those with elevated GPR the proportion with a defined risk factor for fibrosis. (E) Proportion of the population with an elevated AST/ALT ratio, and among those with an elevated ratio the proportion with a self-reported history of alcohol intake. ALT, alanine transaminase; AST, aspartate transaminase; AST/ALT ratio, aspartate/alanine ratio; GGT, gamma-glutamyl transpeptidase; GPC, General Population Cohort; GPR, GGT to platelet ratio; HBV, hepatitis B virus; LFTs, liver function tests; ULN, upper limit of normal.

ULN of AST, the proportion of the population classified as having a score consistent with liver fibrosis changes according to whether the ARR or LRR is used (table 1). Based on previous validation among African individuals, there is some limited evidence to suggest that GPR is the most accurate score for staging liver fibrosis[13]; applying this approach, there is a prevalence of almost one in four adults with liver fibrosis in this population.

### Evidence for the contribution of alcohol to liver disease

The prevalence of AST/ALT ratio >2, suggestive of alcoholic hepatitis, was 11% (888/8099; figure 1E). The median and IQR of GGT among alcohol drinkers were significantly larger than non-drinkers (23.2 (15.6–38.9) vs 17.3 (12.8–23.7); online supplementary table 4[5]). There was a significant relationship between self-reported alcohol consumption and elevated AST/ALT ratio (p<0.001; online supplementary figure 3[5]). However, 57% of participants with AST/ALT ratio >2 reported never having consumed alcohol (figure 1E), possibly

reflecting either under-reporting of alcohol use and/ or other factors that underpin this pattern of LFTs. Self-reported alcohol consumption was associated with raised LFTs, as follows: ALT (adjusted OR 1.33, 95% CI 1.09 to 1.63), AST (adjusted OR 1.53, 95% CI 1.30 to 1.78), GGT (adjusted OR 2.00 95% CI 1.69 to 2.36) and with abnormal fibrosis scores, particularly GPR (adjusted OR 1.96, 95% CI 1.52 to 2.54). All ORs, adjusted ORs, their respective 95% CIs and p values are shown in table 2, and selected variables in figure 2.

A raised GGT level in combination with AST/ALT ratio >2 can be used to increase the sensitivity of detection of alcoholic hepatitis.[9] GGT levels were significantly higher among males with AST/ALT ratio ≥2 (p<0.001), but there was no relationship between GGT and AST/ALT ratio in females (p=0.7); online supplementary figure 4.[5] This potentially indicates that alcohol is of more influence as a cause of an elevated AST/ALT ratio in men than in women. There was no significant association between

**Table 2** Univariate and multivariate analysis for factors associated with abnormal LFTs according to American Reference Range (ARR) for ALT, AST, ALP, GGT and BR, and laboratory markers of fibrosis in adults in the Uganda GPC

| | ALT† OR (95% CI) | AST† OR (95% CI) | ALP †, ‡‡ OR (95% CI) | GGT† OR (95% CI) | BR† OR (95% CI) | FIB-4‡ OR (95% CI) | APRI ‡, § OR (95% CI) | GPR ‡ OR (95% CI) | AST/ALT‡ or (95% CI) | S-index‡, ¶ OR (95% CI) |
|---|---|---|---|---|---|---|---|---|---|---|
| **Univariate analysis** | | | | | | | | | | |
| **Sex** | | | | | | | | | | |
| Male | Ref | Ref | Ref | Ref | Ref | Ref | Ref | Ref | Ref | Ref |
| Female | 2.06 (1.71 to 2.49)*** | 1.04 (0.91 to 1.18)ns | 0.93 (0.84 to 1.01)ns | 1.15 (1.00 to 1.32)* | 0.46 (0.20 to 0.24)*** | 0.64 (0.42 to 0.96)* | 0.38 (0.27 to 0.55)*** | 1.10 (0.89 to 1.38)ns | 0.84 (0.73 to 0.96)* | 0.35 (0.21 to 0.57)*** |
| **Age** | | | | | | | | | | |
| <19 | Ref | Ref | – | Ref | Ref | Ref* | Ref | Ref | Ref | Ref†† |
| 20–29 | 1.33 (1.03 to 1.73)* | 0.9 (0.73 to 1.11)ns | Ref‡‡ | 2.61 (1.92 to 3.56)*** | 1.46 (1.22 to 1.75)*** | | 2.57 (1.41 to 4.71)** | 2.63 (1.72 to 4.03)*** | 0.55 (0.43 to 0.70)*** | |
| 30–39 | 1.58 (1.22 to 2.04)*** | 1.17 (0.95 to 1.43)ns | 0.72 (0.60 to 087)*** | 6.59 (5.00 to 8.72)*** | 1.15 (0.94 to 1.39)ns | | 3.15 (1.76 to 5.68)*** | 6.22 (4.21 to 9.18)*** | 0.67 (0.53 to 0.85)** | |
| 40–49 | 1.41 (1.04 to 1.87)* | 1.47 (1.12 to 1.80)*** | 0.48 (0.38 to 0.59)*** | 8.34 (6.29 to 11.07)*** | 1.02 (0.83 to 1.27)ns | 8.48 (3.95 to 18.18)*** | 4.00 (2.22 to 7.18)*** | 7.63 (5.12 to 11.36)*** | 0.83 (0.65 to 1.05)ns | 5.02 (2.79 to 9.68)*** |
| 50–59 | 1.38 (1.00 to 1.90)* | 1.57 (1.25 to 2.00)*** | 0.82 (0.66 to 1.02)ns | 8.03 (5.93 to 10.86)*** | 0.92 (0.71 to 1.18)ns | 14.60 (9.86 to 31.03)*** | 3.50 (1.80 to 6.73)*** | 9.10 (5.91 to 14.0)*** | 1.11 (0.86 to 1.43)ns | 4.71 (2.31 to 9.59)*** |
| >60 | 1.39 (1.03 to 1.88)* | 1.24 (0.98 to 1.55)ns | 1.28 (1.06 to 1.54)** | 6.84 (5.09 to 9.20)*** | 0.56 (0.42 to 0.74)*** | 34.88 (17.80 to 68.39)*** | 3.68 (2.00 to 7.00)*** | 8.20 (5.42 to 12.41)*** | 2.23 (1.82 to 2.72)*** | 5.43 (2.84 to 10.39)*** |
| **Alcohol** | | | | | | | | | | |
| No | Ref | Ref | Ref | Ref | Ref | Ref | Ref | Ref | Ref | Ref |
| Yes | 1.41 (1.16 to 1.70)*** | 1.57 (1.35 to 1.83)*** | 1.0 (0.86 to 1.13)*** | 2.14 (1.83 to 2.51)*** | 0.99 (0.85 to 1.15)ns | 2.02 (1.22 to 3.32)** | 1.60 (1.04 to 2.31)* | 2.10 (1.61 to 2.66)*** | 1.28 (1.08 to 1.50)** | 6.09 (3.16 to 11.72)*** |
| **BMI§§** | | | | | | | | | | |
| Normal | Ref | Ref | Ref | Ref | Ref | Ref | Ref | Ref | Ref | Ref |
| Underweight | 1.41 (1.12 to 1.77)** | 1.45 (1.23 to 1.71)*** | 1.17 (0.96 to 1.44)ns | 1.42 (1.16 to 1.73)** | 0.69 (0.57 to 0.83)*** | 1.78 (1.06 to 3.00)ns | 1.78 (1.10 to 2.60)* | 1.07 (0.78 to 1.50)ns | 1.62 (1.37 to 1.92)*** | 1.87 (1.04 to 3.33)* |
| Overweight | 1.10 (0.85 to 1.41)ns | 0.73 (0.58 to 0.92)** | 0.93 (0.77 to 1.13)ns | 1.36 (1.11 to 1.66)** | 0.75 (0.59 to 0.95)* | 0.74 (0.35 to 1.56)ns | 0.91 (0.50 to 1.65)ns | 1.15 (0.82 to 1.60)ns | 0.57 (0.42 to 0.76)*** | 0.87 (0.38 to 2.03)ns |
| **HIV status** | | | | | | | | | | |
| Negative | Ref | Ref | Ref | Ref | Ref | Ref | Ref | Ref | Ref | Ref |
| Positive | 1.63 (1.24 to 2.15)*** | 2.30 (1.87 to 2.83)*** | 1.47 (1.19 to 1.81)*** | 4.83 (3.98 to 5.85)*** | 0.21 (0.14 to 0.33)*** | 0.28 (0.07 to 1.20)ns | 1.30 (0.68 to 2.30)ns | 3.88 (2.62 to 5.73)*** | 1.06 (0.80 to 1.42)ns | 4.00 (2.08 to 7.69)*** |
| **HBV status** | | | | | | | | | | |

Continued

**Table 2** Continued

| | ALT† OR (95% CI) | AST† OR (95% CI) | ALP †, ‡‡, OR (95% CI) | GGT† OR (95% CI) | BR‡ OR (95% CI) | FIB-4‡ OR (95% CI) | APRI‡, § OR (95% CI) | GPR ‡ OR (95% CI) | AST/ALT‡ or (95% CI) | S-index‡, ¶ OR (95% CI) |
|---|---|---|---|---|---|---|---|---|---|---|
| Negative | Ref | Ref | Ref | Ref | Ref | Ref | Ref | Ref | Ref | Ref |
| Positive | 2.61 (1.77 to 3.84)*** | 2.52 (1.84 to 3.44)*** | 1.07 (0.72 to 1.60)ns | 1.80 (1.24 to 2.60)*** | 1.10 (0.76 to 1.60)ns | 2.01 (0.62 to 6.50)ns | 3.56 (1.80 to 7.10)*** | 4.24 (2.27 to 7.93)*** | 0.98 (0.63 to 0.15)ns | 4.92 (2.07 to 11.69)*** |
| **Multivariate analysis** | | | | | | | | | | |
| Sex | | | | | | | | | | |
| Male | Ref | Ref | Ref | Ref | Ref | Ref | Ref | Ref | Ref | Ref |
| Female | 2.30 (1.89 to 2.81)*** | 1.20 (1.04 to 1.38)* | 2.11 (1.83 to 2.44)*** | 1.01 (0.86 to 1.19)ns | 0.46 (0.40 to 0.53)*** | 0.62 (0.40 to 0.97)* | 0.42 (0.30 to 0.62)*** | 1.11 (0.87 to 1.41)ns | 0.90 (0.78 to 1.06)ns | 0.37 (0.22 to 0.63)*** |
| Age | | | | | | | | | | |
| <19 | Ref | Ref | – | Ref | Ref | Ref†† | Ref | Ref | Ref | Ref†† |
| 20–29 | 1.26 (0.95 to 1.68)ns | 0.89 (0.70 to 1.12)ns | Ref‡‡ | 1.69 (1.19 to 2.41)** | 1.52 (1.25 to 1.84)*** | | 3.22 (1.66 to 6.22)** | 1.86 (1.19 to 2.92)** | 0.57 (0.44 to 0.75)*** | |
| 30–39 | 1.35 (1.00 to 1.80)* | 1.00 (0.79 to 1.27)ns | 0.68 (0.56 to 0.82)*** | 3.96 (2.87 to 5.46)*** | 1.29 (1.02 to 1.59)* | | 3.55 (1.81 to 7.00)*** | 3.70 (2.43 to 5.66)*** | 0.72 (0.55 to 0.95)* | |
| 40–49 | 1.13 (0.83 to 1.56)ns | 1.20 (0.95 to 1.52)ns | 0.46 (0.37 to 0.57)*** | 4.87 (3.54 to 6.70)*** | 1.17 (0.94 to 1.47)ns | 7.04 (3.19 to 15.52)*** | 4.00 (2.04 to 7.82)*** | 4.45 (2.88 to 6.87)*** | 0.93 (0.71 to 1.21)ns | 2.68 (1.37 to 5.26)** |
| 50–59 | 1.09 (0.77 to 1.55)ns | 1.29 (0.99 to 1.67)ns | 0.82 (0.66 to 1.02)ns | 5.02 (3.58 to 7.02)*** | 1.01 (0.78 to 1.32)ns | 11.29 (5.13 to 24.80)*** | 3.45 (1.65 to 7.22)** | 5.75 (3.61 to 9.15)*** | 1.22 (0.92 to 1.61)ns | 2.76 (1.29 to 5.90)** |
| >60 | 1.13 (0.81 to 1.57)ns | 1.00 (0.78 to 1.30)ns | 1.32 (1.09 to 1.59)** | 4.98 (3.59 to 6.90)*** | 0.60 (0.45 to 0.80)*** | 25.15 (12.32 to 51.35)*** | 3.50 (1.73 to 7.11)** | 5.39 (3.42 to 8.47)*** | 2.20 (1.74 to 2.77)*** | 3.34 (1.63 to 6.84)*** |
| Alcohol | | | | | | | | | | |
| No | Ref | Ref | – | Ref | – | Ref | Ref | Ref | Ref | Ref |
| Yes | 1.33 (1.09 to 1.63)** | 1.53 (1.30 to 1.78)*** | – | 2.00 (1.69 to 2.36)*** | – | 2.05 (1.24 to 3.40)** | 1.51 (1.00 to 2.27)* | 1.96 (1.52 to 2.54)*** | 1.26 (1.06 to 1.50)** | 5.23 (2.72 to 10.04)*** |
| BMI‡ | | | | | | | | | | |
| Normal | Ref | Ref | – | Ref | Ref | – | Ref | – | Ref | – |
| Underweight | 1.40 (1.11 to 1.75)** | 1.44 (1.21 to 1.70)*** | – | 1.37 (1.11 to 1.68)** | 0.70 (0.58 to 0.83)*** | – | 1.72 (1.11 to 2.65)* | – | 1.61 (1.36 to 1.91)*** | – |
| Overweight | 1.12 (0.87 to 1.44)ns | 0.75 (0.60 to 0.95)* | – | 1.47 (1.19 to 1.82)*** | 0.72 (0.57 to 0.92)** | – | 0.95 (0.52 to 1.73)ns | – | 0.56 (0.42 to 0.76)*** | – |
| HIV status | | | | | | | | | | |
| Negative | Ref | Ref | Ref | Ref | Ref | – | – | Ref | – | Ref |
| Positive | 1.59 (1.20 to 2.10)*** | 2.13 (1.72 to 2.63)*** | 1.47 (1.19 to 1.81)*** | 4.76 (3.89 to 5.82)*** | 0.22 (0.14 to 0.34)*** | – | – | 3.84 (2.58 to 5.70)*** | – | 3.58 (1.84 to 6.94)*** |

Continued

**Table 2** Continued

| | ALT† OR (95% CI) | AST† OR (95% CI) | ALP †, ‡‡, OR (95% CI) | GGT† OR (95% CI) | BR† OR (95% CI) | FIB-4‡ OR (95% CI) | APRI ‡, § OR (95% CI) | GPR ‡ OR (95% CI) | AST/ALT‡ or (95% CI) | S-index‡, ¶ OR (95% CI) |
|---|---|---|---|---|---|---|---|---|---|---|
| **HBV status** | | | | | | | | | | |
| Negative | Ref | Ref | – | Ref | – | – | Ref | Ref | – | Ref |
| Positive | 2.61 (1.76 to 3.86)*** | 2.40 (1.74 to 3.31)*** | – | 1.65 (1.11 to 2.45)* | – | – | 3.60 (1.79 to 7.27)*** | 4.26 (2.23 to 8.12)*** | – | 4.37 (1.80 to 10.58)*** |

Significance values: *=(p<0.05), **=(p<0.01), ***=(p<0.001), ns=(p>0.05).
†Abnormal LFTs, according to ARR, are defined as test results outside of the following ranges: ALT (male: 10–55 U/L, female: 7–30U/L), AST (male: 10–40 U/L, female: 9–32U/L), GGT (male: 8–61 U/L, female: 5–36 U/L), BR (0–17 mmol/L), ALP (male: 45–115U/L, female: 30–100 U/L).
‡Threshold used to predict liver fibrosis: APRI > 0.7. FIB-4 >3.25. GPR >0.32. RPR >0.825. S-index >0.3
§APRI score calculated using ULN (upper limit of normal) of AST using African reference range
¶An S-index score of >0.3 is suggestive of liver fibrosis.
††Reference age group consists of all individuals under the age of 39
‡‡Individuals under the age of 19 were excluded. Reference age group is 20 – 29
§§BMI classification according to WHO (weight/height²: kg/m²): Underweight (<18.5 kg/m²), Normal weight (18.5 – 24.99 kg/m²), Overweight (25.0 – 29.99 kg/m²), Obese (>30.0 kg/m².
ALP, alkaline phosphatase; ALT, alanine transaminase; APRI, AST to Platelet Ratio Index; ARR, American reference range; AST, aspartate transaminase; AST/ALT ratio, aspartate/alanine ratio; BMI, body mass index; BR, total bilirubin; FIB-4, fibrosis 4; GGT, gamma-glutamyl transpeptidase; GPC, General Population Cohort; GPR, GGT to platelet ratio; HBV, hepatitis B virus; LFTs, liver function tests; LRR, local reference range; RPR, red cell distribution width to platelet ratio; ULN, upper limit of normal.

AST/ALT ratio ≥2 and the presence of an elevated GPR score, predicting fibrosis (p=0.2; data not shown). We calculated PAR as a way to assess the relative contribution of different risk factors to the overall burden of liver disease; table 3. Overall, the most striking contribution arose from reported alcohol consumption, which accounted for 64% of abnormal S-index scores, 32% of elevated FIB-4 scores and 19% of GPR abnormalities.

### Abnormal LFTs and/or elevated fibrosis scores are associated with sex, age and body mass index (BMI)

Compared with males, females were less likely to have high fibrosis scores based on FIB-4 (adjusted OR: 0.6), APRI (adjusted OR: 0.42) and S-index (adjusted OR: 0.37). FIB-4 score increased markedly with age: adults aged 40–49 (adjusted OR: 7.04), 50–59 (adjusted OR: 11.29) and adults >60 years (adjusted OR: 25.15) were more likely to have a higher FIB-4 than individuals <40 years. Elevated BMI was associated only with a rise in GGT (adjusted OR: 1.47). However, being underweight was associated with a more pronounced pattern of liver derangement, including elevations in ALT (adjusted OR: 1.40), AST (adjusted OR: 1.44), GGT (adjusted OR: 1.37), abnormal fibrosis scores (APRI, adjusted OR: 1.72) and with raised AST/ALT ratio (adjusted OR: 1.61). 95% CIs in each case are shown in table 2.

### Relationship between BBV infection and liver disease

HIV infection was associated with abnormal LFTs, with significant OR for increased ALT, AST, ALP and GGT, as well as with raised GPR and S-index (on univariate and multivariate analysis; table 2). Individuals with HIV or HBV infection had higher LFTs (ALT, AST, ALP, GGT) and elevated liver fibrosis scores (FIB-4, APRI, GPR and S-index) compared with uninfected individuals (online supplementary table 4[5]). HBV infection was significantly associated with a rise in hepatic transaminases (adjusted OR for raised ALT and AST 2.6 and 2.4, respectively), and with liver fibrosis as measured by APRI and GPR (adjusted OR 3.6 and 4.2, respectively). We investigated the prevalence of BBV infection among individuals with raised fibrosis scores. There was an association between the presence of HIV or HBV and raised GPR (p=0.005) and S-index (p<0.001). HIV and HBV were associated with a lesser proportion of liver disease than alcohol based on calculation of PAR (table 3), but still contributed to elevations in both LFTs and fibrosis scores. The OR for deranged LFTs/fibrosis scores in the context of HIV or HBV infection is shown in figure 2.

### Liver disease of unknown aetiology

Among individuals with GPR>0.32, 33.8% had either BBV infection or had AST/ALT ratio >2 (suggesting potential alcoholic hepatitis; figure 1D; online supplementary figure 5[5]). However, this illustrates that 66% have raised fibrosis scores in the absence of a history of alcohol use, or HIV or HBV infection, suggesting that other factors unaccounted for in this study are likely to be contributing

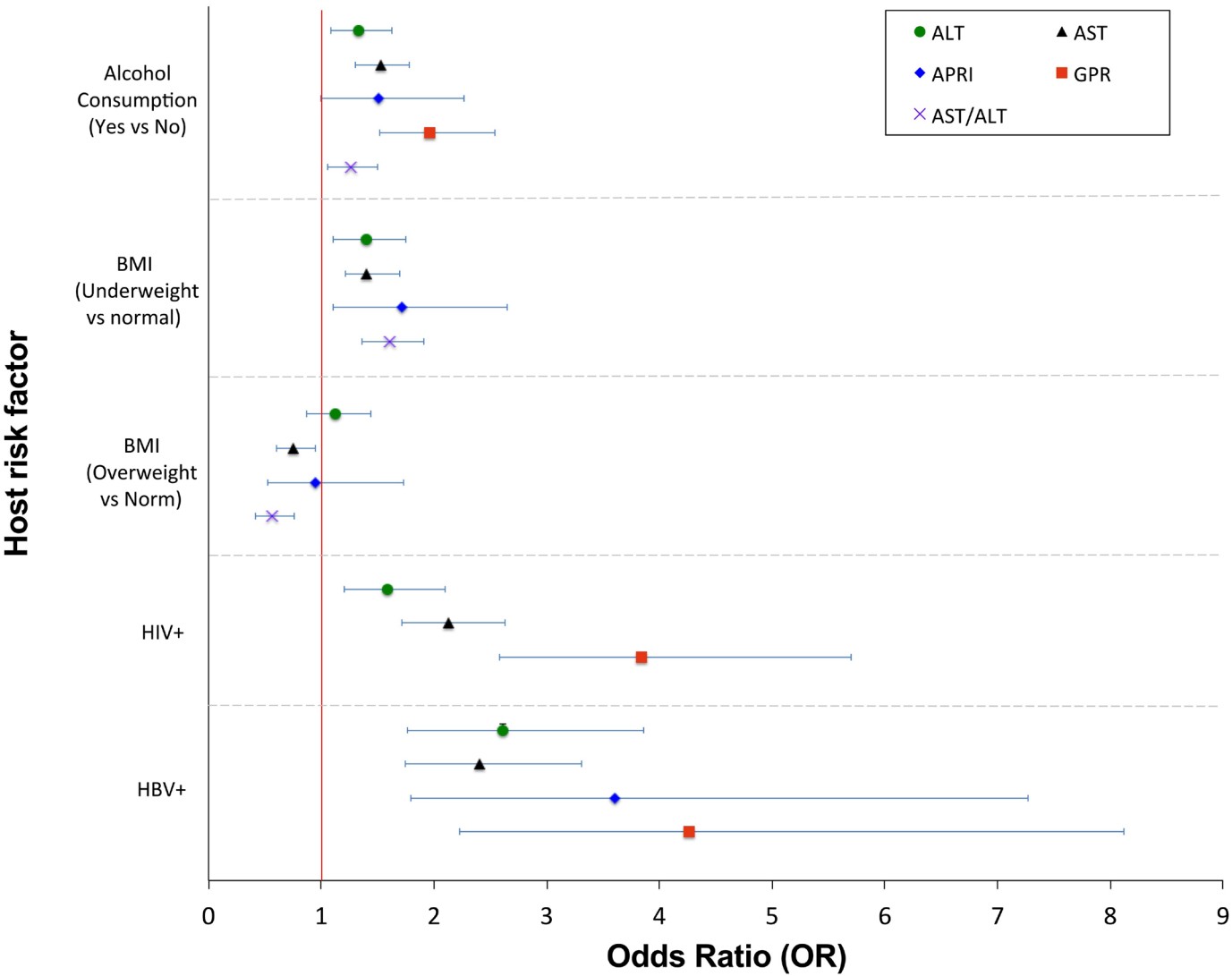

**Figure 2** Forest plots to show OR for host risk factors and elevated LFTs or fibrosis scores in the Uganda GPC. Data are presented for the final multivariate model for ALT, AST, APRI, GPR and AST/ALT, showing variables that were independently associated with the outcome (statistically significant at the p<0.05 level after adjusting for other variables). ALT, alanine transminase; APRI, AST to Platelet Ratio Index; AST, aspartate transminase; AST/ALT, aspartate/alanine ratio; BMI, body mass index; GPC, General Population Cohort; GGT, gamma-glutamyl transpeptidase; GPR, GGT to platelet ratio; LFTs, liver function tests.

to the overall burden of liver disease. In the setting of a population-based cohort (where the background prevalence of liver fibrosis is relatively low), many of those with an abnormal test result may not have liver disease; these 'false positive' cases of elevated GPR may also account for some of the 66% in whom we could not identify a risk factor. True prevalence of liver disease cannot be ascertained until reference ranges have been more carefully defined, correlating LFTs and fibrosis scores with the confirmed presence of underlying liver disease based on imaging or biopsy.

## DISCUSSION

Liver disease is not well characterised in many parts of sSA despite the high prevalence of HIV and HBV, and potential exposure to hepatotoxins.[1 3] In this study, we used cross-sectional data from a large population cohort to estimate the burden of liver disease and to assess the possible impact of BBV infection and alcohol consumption. The prevalence of abnormal LFTs depends on the reference range that is applied. The ARR suggests a higher prevalence of liver disease, therefore including more false-positives. The LRR was established based on individuals recruited from several countries across Africa (Rwanda, Uganda, Kenya, Zambia).[18] While the values were derived from purportedly healthy adults, it is impossible to rule out a high background prevalence of underlying liver disease; in defining higher values for the ULN of all tests, the LRR is more susceptible to false-negatives if used to screen for liver disease. Composite fibrosis scores have been developed with the aim of improving sensitivity of detection of liver disease,[32] but these it is striking

**Table 3** Relative risk, Population Attributable Risk (PAR) and the number of individuals with abnormal LFTs in the Uganda GPC.

| Variable | ALT * | AST * | ALP* | GGT * | BR* | Fib-4† | APRI†, ‡ | GPR† | AST/ALT† | S-index†, § |
|---|---|---|---|---|---|---|---|---|---|---|
| **Alcohol¶** | | | | | | | | | | |
| Abnormal result n (%) | 248 (8.5) | 467 (16.0) | 533 (19.6) | 555 (19) | 381 (13.1) | 72 (11.0) | 80 (12.25) | 260 (39.8) | 379 (13.0) | 60 (9.2) |
| RR (95% CI) | 1.4 (1.2–1.6) | 1.5 (1.4–1.7) | 1.2 (0.9–1.7) | 2.9 (2.6–3.4) | 1.0 (0.9–1.1) | 5.0 (3.2–7.7) | 2.3 (1.7–3.2) | 2.7 (2.3–3.2) | 1.3 (1.2–1.5) | 8.7 (4.8–15.6) |
| PAR (%)** | 11.3% | 15.9% | 0.6% | 41.3% | 0.3% | 58.2% | 31.3% | 37.1% | 10.8% | 72.7% |
| Adjusted PAR (%)**†† | 10.0% | 13.9% | −2.6% | 26.7% | 1.0% | 32.4% | 16.2% | 19.4% | 8.0% | 64.0% |
| **HIV*** | | | | | | | | | | |
| Abnormal result n (%) | 71 (11.7) | 144 (23.7) | 142 (24.8) | 227 (37.3) | 21 (3.5) | ‡(1.6) | 14 (11.0) | 73 (57.5) | 59 (9.7) | 15 (11.8) |
| RR (95% CI)† | 1.7 (1.4–2.2) | 2.0 (1.8–2.4) | 1.2 (1.1–1.4) | 4.2 (3.7–4.8) | 0.3 (0.2–0.4) | 0.3 (0.1–1.1) | 1.5 (0.9–2.5) | 2.7 (2.3–3.3) | 0.9 (0.7–1.1) | 3.6 (2.1–6.1) |
| PAR (%)** | 5.3% | 7.3% | 2.2% | 19.5% | −6.0% | −5.09% | 3.1% | 10.5% | −0.9% | 14.7% |
| Adjusted PAR (%)**†† | 4.3% | 6.5% | 1.1% | 17.6% | −6.0% | −4.6% | 1.4% | 8.3% | −0.1% | 13.6% |
| **HBV¶** | | | | | | | | | | |
| Abnormal result n (%) | 33 (15.0) | 56 (25.4) | 32 (19.5) | 39 (17.7) | 35 (16) | ¶(8.2) | 13 (26.53) | 25 (51.0) | 22 (10.0) | 8 (16.3) |
| RR (95% CI) | 2.2 (1.6–3.0) | 2.1 (1.7–2.7) | 0.9 (0.7–1.3) | 1.6 (1.2–2.2) | 1.2 (0.9–1.7) | 1.6 (0.6–4.1) | 1.5 (0.9–2.5) | 2.2 (1.7–3.0) | 0.9 (0.6–1.4) | 4.6 (2.3–9.0) |
| PAR (%)** | 3.1% | 2.9% | −0.2% | 1.7% | 0.6% | 1.5% | 3.1% | 3.1% | −0.2% | 8.6% |
| Adjusted PAR (%)**†† | 3.3% | 2.8% | 0.02% | 1.4% | 0.2% | 1.4% | 5.7% | 2.9% | −0.3% | 7.6% |

Analysis was done according to ARR for ALT, AST, ALP, GGT, and BR.
*Number of abnormal result, RR and PAR (%) are based on individuals who were classified as positives within each variable (ie, alcohol drinkers, HIV positive, HBV positive).
†Threshold used to predict liver fibrosis: APRI>0.7, FIB-4 >3.25, GPR >0.32, RPR >0.825 and S-index >0.3.
‡APRI score calculated using ULN of AST using African reference range.
§An S-index score of >0.3 is suggestive of liver fibrosis
¶ number of abnormal result, RR and PAR (%) are based on individuals who were classified as positives within each variable (ie. Alcohol drinkers, HIV positive, HBV positive)
**A measure of 0 indicates of no association between the risk factor and abnormal LFTs. A positive value indicates that the exposure to the risk factor is a risk factor, while a negative value indicates that it is a protective factor.
††Adjusted for age, sex, alcohol consumption, HBV diagnosis, HIV status, and BMI.
ALP, alkaline phosphatase; ALT, alanine transaminase; APRI, AST to Platelet Ratio Index; ARR, American reference range; AST, aspartate transaminase; AST/ALT ratio, aspartate/alanine ratio; BMI, body mass index; BR, total bilirubin; FIB-4, fibrosis 4; GGT, gamma-glutamyl transpeptidase; GPR, GGT to platelet ratio; HBV, hepatitis B virus; LFTs, liver function tests; PAR, population attributable risk; ULN, upper limit of normal.

that there is a large variation in the prevalence of liver fibrosis estimated by different scores, ranging from 23.5% based on assessment using GPR, down to <1% with RPR. This discrepancy highlights the differing performance of different scores, but in the absence of elastography data, we are currently unable to determine which test offers the most accurate assessment.

LFTs are a blunt tool for assessment of liver health, with many potential confounding factors. This current study only accounts for a limited range of aetiological agents, and we did not include other potentially relevant factors such as Schistosomiasis infection, exposure to aflatoxin and use of traditional medications. Furthermore, LFTs were measured at only one point in time, potentially over-calling liver disease as a result of transient abnormalities. Further studies will be required to investigate a greater range of risk factors, and to undertake longitudinal follow-up.

Fibrosis scores also depend on platelet count which can be influenced by diverse factors. For example, in some African populations, thrombocytopenia is common due to infections such as malaria, schistosomiasis, HIV or endemic parasites, as well as being influenced by inflammatory conditions and certain drugs.[10 11] We only

had platelet counts for a subset of our study population, limiting the number for whom we could determine APRI, FIB-4, GPR, S-index and RPR scores. Data surrounding the use of these scores in sSA is variable, but since in many low-income settings alternative diagnostic equipment is unavailable, non-invasive approaches are vital to estimate liver damage and to stratify clinical management decisions. The finding that almost 1:4 individuals in this population study had an abnormal GPR score is concerning and striking. This could be influenced by high GGT values (potentially in association with alcohol), or low platelet counts (for the reasons outlined above). However, it should also be noted that we used stringent thresholds for GGT, with different thresholds for the ULN in males and females (online supplementary table 1[5]), which influence the proportion of the population meeting the threshold for elevation of both GGT and GPR.

APRI and FIB-4 are currently recommended by the WHO for assessment of hepatic fibrosis in patients with chronic HBV or HCV infection.[33 34] However, the evidence is limited, and to some extent conflicting. One report concludes that APRI is more accurate in assessing liver fibrosis among individuals with chronic HCV compared

with HBV infection.[12] Meanwhile, GPR and S-index have been validated in small studies in sSA, and have been associated with improved classification of liver fibrosis in chronic HBV infection when compared with APRI and FIB-4.[13–15] A study in Ethiopia reported a similar specificity of APRI, GPR and FIB-4 for the detection of fibrosis and cirrhosis.[15] It is apparent that either larger studies, or indeed a meta-analysis, are required to further assess the accuracy of these tests in different populations and in the context of different underlying disease processes. GPR and S-index may be worthwhile options to include in routine clinical practice to assess for liver fibrosis in African populations, given the high burden of HBV in this continent.[35 36] RPR has been used to detect fibrosis among individuals with chronic HBV in China;[28] however, this score was excluded from our analysis due to a very small number of individuals falling above the suggested threshold for fibrosis.

The prevalence of AST/ALT ratio >2 in this population is 11%, suggesting potential alcoholic hepatitis,[37] concordant with a previous study in Uganda in which 10% of the population was estimated to have alcoholic hepatitis,[38] and with data from Uganda's non-communicable diseases risk factor survey which estimated that almost 10% of Ugandan adults have alcohol use disorders.[39] Data from emergency attendances at the Mulago Hospital in Kampala recorded 47% who reported alcohol use, while 21% and 10% met the study definitions of alcoholic misuse and alcoholic liver disease, respectively.[38] Our data are based on self-reported alcohol consumption so may underestimate the true extent of alcohol use. We were unable to quantify alcohol intake or the nature of the alcohol consumed: this is challenging as alcohol is often home-brewed or home-distilled from locally grown grains or fruits, and the alcohol content may vary widely; for exmaple, the alcohol content of locally produced maize-based brews and liquor in Kenya ranged from 2% to 7% and 18% to 53%, respectively.[39] The global challenge of morbidity and mortality associated with alcohol use is highlighted by recent studies from the Global Burden of Disease consortium, in which alcohol ranks as the seventh highest cause of disability-adjusted life years (DALYs) and deaths, and worldwide[2] - together with HBV infection - is a leading aetiological agent of liver cancer.[40] Further data collection using validated tools to quantify the frequency, volume and patterns of alcohol consumption will be important to improve insights into the relationship between alcohol and liver disease in our population setting.

The calculation of PAR that we have undertaken in this study should be interpreted with caution, as we recognise that robust assessment of exposure to alcohol is difficult, and the markers we are using to represent underlying liver disease each comes with associated caveats. We have nevertheless included this analysis as part of our output on the grounds that it is congruent with other aspects of the analysis in highlighting a likely significant role for alcohol as a driver of liver disease, and therefore may be of influence in informing future studies as well as underpinning appropriate interventions.

Abnormal LFTs are common in HIV infection for diverse reasons including direct cytopathic effects of HIV on hepatocytes, coinfection with other BBVs, opportunistic infection, malignancy, antiretroviral therapy (ART) or other drugs, or secondary to other factors such as alcoholism.[41–44] Although a proportion of our study population with fibrosis were infected with BBV (21.6%) and/or had a history of alcohol consumption (12.2%), there was a residual proportion with scores suggestive of fibrosis and AST/ALT ratio >2 who cannot be accounted for through either alcohol or BBV infection. This is in keeping with other studies from Africa that report a high proportion of cases of liver disease that are not attributable to viral infection or alcohol and could be as a result of other understudied factors such as NAFLD and use of traditional medicine.[38 45] Khat chewing (a popular recreational drug in some settings) was recently found to be a major cause of unexplained liver disease in east Ethiopia.[45] Aflatoxin exposure is associated with liver cirrhosis and is among the major causes of hepatocellular carcinoma globally, with most cases reported from sSA. Within a previous study of the GPC, >90% of individuals had evidence of exposure.[46–48]

In our population women were significantly more likely to be overweight than men. This may be associated with a higher incidence of NAFLD in women. However, typically only mild rises in ALT are seen, and 80% of those with NAFLD have normal LFTs,[49–51] so may not be identified within our current dataset. Diagnosis of NAFLD therefore depends on ultrasound scan; previous studies have consistently shown 70%–80% of obese patients have NAFLD on imaging.[50 52 53] These imaging modalities were not available in our population, so we are unable to comment specifically on the possible prevalence of NAFLD. Interestingly, in this setting low body weight was more associated with deranged LFTs and with biochemical evidence of liver fibrosis, suggesting a range of pathology that may contribute to liver disease, including organ-specific effects of undernutrition or stunting,[40] as well as the effect of general systemic illness. Further studies are required to investigate the specific relationship between BMI and liver fibrosis in African populations.

In African populations, HCV infection has frequently been often over-reported due to a reliance on HCV-antibody (HCV-Ab) testing, which detects not only current infection but also previous exposure, and is known to be susceptible to false positive results.[30] In this cohort, 298 (3.7%)/8145 individuals tested HCV-Ab positive, but among these only 13 were HCV RNA positive (overall prevalence 13/8145=0.2%).

Appropriate reference ranges for LFTs are necessary to contribute to an understanding of the burden and aetiology of liver disease. Further work is required to determine appropriate thresholds for the ULN of different parameters in different settings in sSA, and to determine which fibrosis score is most specific, through application

of a more widespread approach to elastography and/or other imaging. At present, we have identified alcohol, HIV and HBV as risk factors for deranged LFTs and elevated liver fibrosis scores, with a particularly striking contribution made by alcohol, but further investigation is needed to determine other risk factors that contribute to liver disease in this setting.

**Author affiliations**

[1]Faculty of Infectious and Tropical Diseases, London School of Hygiene and Tropical Medicine, London, UK
[2]Nuffied Department of Medicine, University of Oxford, Oxford, UK
[3]Medical Research Council/Uganda Virus Research Institute, Entebbe, Uganda
[4]Department of Microbiology and Infectious Diseases, Oxford University Hospitals NHS Foundation Trust, Oxford, UK
[5]Department of Global Health & Development, London School of Hygiene and Tropical Medicine, London, UK
[6]NIHR BRC, John Radcliffe Hospital, Oxford, UK
[7]Department of Health Sciences, University of York, York, UK

**Contributors** GO, JS, PM and RN conceived the study. AK, GA, JS and RN helped in data collection. JM, JPH, LOD, ALM and PM analysed the data. GO, JM, JPH, LOD, PM and RN wrote the manuscript. All authors helped in revising the manuscript and have read and approved the manuscript.

**Funding** The General Population Cohort is jointly funded by the UK Medical Research Council (MRC) and the UK Department for International Development (DFID) under the MRC/DFID Concordat agreement. The work on liver function also received additional funding from the MRC (grant numbers G0801566 and G0901213-92157). JM is funded by a Leverhulme Mandela Rhodes Scholarship. PM is funded by the Wellcome Trust (grant number 110110). LOD is funded by NIHR.

**Competing interests** None declared.

**Patient and public involvement** Patients and/or the public were not involved in the design, or conduct, or reporting, or dissemination plans of this research.

**Patient consent for publication** Not required.

**Ethics approval** Ethics approval was provided by the Science and Ethics Committee of the Uganda Virus Research Institute (GC/127/12/11/06), the Ugandan National Council for Science and Technology (HS870) and the East of England-Cambridge South (formerly Cambridgeshire 4) NHS Research Ethics Committee UK (11/H0305/5). All participants provided written informed consent.

**Provenance and peer review** Not commissioned; externally peer reviewed.

**Data availability statement** Data are available in a public, open access repository. All data relevant to the study are included in the article or uploaded as supplementary information. All data generated or analysed during this study are included in this published article, and its Supplementary Information files, which are accessible on-line at Figshare: https://doi.org/10.6084/m9.figshare.8292194.

**ORCID iDs**
Jeffrey P Hau http://orcid.org/0000-0002-3656-6538
Philippa C Matthews http://orcid.org/0000-0002-4036-4269

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
