## [Reviewer comments · BMJ Open]

ARTICLE DETAILS

TITLE (PROVISIONAL)	Liver function tests and fibrosis scores in a rural population in Africa: a cross-sectional study to estimate the burden of disease and associated risk factors
AUTHORS	O'Hara, Geraldine; Mokaya, Jolynne; Hau, Jeffrey; Downs, Louise; McNaughton, Anna; Karabarinde, Alex; Asiki, G; Seeley, Janet; Matthews, Philippa; Newton, Robert

VERSION 1 – REVIEW

REVIEWER	Mark Sonderup University of Cape Town, Medicine
REVIEW RETURNED	17-Nov-2019

GENERAL COMMENTS	Thank you for allowing me to review this very well presented paper. It adds to a limited body of knowledge on the extent of liver health in sub Saharan Africa. I have a few comments/points of clarity: 1. The usefulness of the cut-off values for APRI in patients with HBV infection have been questioned. Authors may wish to mention this in their introduction. Although the study cannot assess this, do the authors have any data from their cohort to support the suggestion that APRI rule-in and rule scores for cirrhosis need altering for chronic HBV infection?2. Given the high HIV rates (understandably) and the issue that lower platelet counts are a relatively frequent observation in patients with HIV infection, did the authors look at APRI and other markers of fibrosis in HIV positive versus HIV negative patients?3. Do the authors find the GPR rate of advanced fibrosis of 25% (1:4) alarmingly high - is this perhaps an overestimation? Are there any other suggested reasons?
---

REVIEWER	Asgeir Johannessen Oslo University Hospital Centre for imported and tropical diseases Oslo Norway
REVIEW RETURNED	27-Nov-2019

GENERAL COMMENTS	This well-written study explored LFTs among a rural population in Uganda. The authors calculated various scores of liver fibrosis and tried to link this to demographic and other population characteristics. Since data is scarce on the performance of these liver fibrosis scores from rural Africa, the paper provides new and relevant information. The lack of other measures of liver fibrosis, however, is a clear limitation.
---

	Specific comments:  • Page 9, line 13-17: The most striking result is the very large variation in fibrosis scores, from 23.5% with GPR to 0.1% with RPR. I think this deserves a comment in the Discussion. • Page 9, line 13-17: In the original paper by Lemoine et al, the GPR was based on a GGT threshold of 58 in both men and women, whereas you employ a much lower cutoff for women. This should be commented (in the Discussion) since it automatically results in a higher GPR result which can explain why you find a higher proportion above the threshold using this test. • Page 9, line 24-28 (“Based on previous validation among African...”): This is very uncertain, since results are scarce and to a certain degree conflicting. See for example Desalegn et al, Liver Int 2017, where APRI, GPR and FIB-4 were quite similar in performance. • Page 9, Alcohol paragraph: Don’t you have information about quantity of alcohol consumed? The categories you report are not very helpful in assessing whether these people have a harmful alcohol consumption, and thereby might have alcoholic hepatitis. You write in the discussion that the quantity of alcohol in local drinks is hard to estimate, but perhaps you have recorded how often they drink so that you can make a rough estimate? E.g. daily drinkers vs. the rest? • Page 9-10: PAR: I think you overstretch your data here. With so blunt tools at hand, you should be very cautious about making a statement about PAR for any risk factor. I think the article is better if you skip it. You risk diluting your nice and valuable results with these very speculative (and probably erratic) PAR results. • Table 2: This table is hard to use for others who are interested in the subject. I suggest that you keep the same format, but instead of giving OR for abnormal test results (which is not very interesting) you can give the median result with interquartile range for each specific test. For example, for APRI you give the absolute APRI result for men and women, and then you can highlight whether the difference is statistically significant. By doing this, others can see the absolute APRI results and make up an idea of normal values in the different population groups. • Page 10, “Abnormal LFTs...” paragraph: You repeat “compared to males” twice in the first sentence. • Page 10, “Relationship between BBV...” paragraph (and suppl fig 5): HIV infection in itself can give low platelets, and therefore all the fibrosis markers you investigate can be erratic in this group (see for example Johannessen A, Lemoine M. Liver Int 2015: 2059). And HIV treatment can give elevated LFTs, which means that these tests might be less accurate (or even useless?) in this population. Thus, your conclusion “Therefore, GPR and S-index may be the most sensitive markers of inflammation...” is not correct. I see that you address this in the Discussion, but you should avoid making this statement here. • Page 10-11, Liver disease of unknown etiology chapter: Here you seem to assume that GPR (and the other NIVs) have 100% positive predictive value. But in a population-based cohort like yours, where the background prevalence of liver fibrosis is low, the majority of those with an abnormal test result will NOT have liver disease. You should acknowledge this also in the Results chapter. • Page 13, 1st paragraph: You should also mention khat chewing (which is not a traditional medicine, but a very popular recreational drug), which was recently found to be a major cause of unexplained liver disease in east Ethiopia (ref: Orlien S, et al. Hepatology 2018: 248-257). And another study by the same group
--	---

	found that more than half of the liver disease in the area was not caused by alcohol or viral agents (Orlien S, et al. BMC Gastroenterol 2018;18:27). • Discussion, last paragraph: In the last sentence you should change “liver fibrosis” to “liver fibrosis tests” since you provide only indirect evidence of liver fibrosis.
REVIEWER	CW Spearman Division of Hepatology, Department of Medicine, Faculty of Health Sciences, U of Cape Town
REVIEW RETURNED	27-Nov-2019
GENERAL COMMENTS	This is a well written manuscript which provides further important insight into the need to assess the validity of fibrosis scores in Africa and the appropriate normal reference ranges for liver function tests in Africa.

VERSION 1 – AUTHOR RESPONSE

Reviewer(s)' Comments to Author:

Reviewer: 1 (Mark Sonderup, University of Cape Town)

Thank you for allowing me to review this very well presented paper. It adds to a limited body of knowledge on the extent of liver health in sub Saharan Africa. Thank you for positive feedback

I have a few comments/points of clarity:

1. The usefulness of the cut-off values for APRI in patients with HBV infection have been questioned. Authors may wish to mention this in their introduction. Although the study cannot assess this, do the authors have any data from their cohort to support the suggestion that APRI rule-in and rule scores for cirrhosis need altering for chronic HBV infection?
We have added to the introduction the specific point about problems with thresholds for APRI scores, saying: ‘GPR has recently been reported as an independent predictor of significant fibrosis in treatment naïve Gambian patients with chronic hepatitis B (CHB) infection, while the usefulness of cut-off values for APRI scores in CHB has been questioned’, with a new reference to support this statement (doi: 10.1111/jvh.13246) We have also made amendments (also based on specific feedback from reviewer 3) to the discussion to highlight the extent to which the existing literature is limited and inconsistent with respect to the evaluation and performance of different fibrosis scores (these specific changes are presented in more detail in our response to reviewer 3, below).
We completely agree about the importance of determining APRI rule in/rule out scores for cirrhosis in the context of chronic HBV infection. The discrepancies between the proportions with abnormal APRI vs other fibrosis scores certainly suggest differing performance between tests, but in the absence of elastography data we do not know which test offers the most accurate assessment; we have added this as a final sentence to the first paragraph of the discussion section. The reviewer is right to say we cannot address this question further based on the current dataset (but might be interested to know that we are planning to move on to assess elastography scores in this setting so that we can address this question in future).

2. Given the high HIV rates (understandably) and the issue that lower platelet counts are a relatively frequent observation in patients with HIV infection, did the authors look at APRI and other markers of fibrosis in HIV positive versus HIV negative patients?

We have undertaken this analysis, with the results presented in the new Suppl Table 4 (as suggested by Reviewer 2, comment pertaining to Table 2). APRI, GPR and FIB-4 and S index are all significantly elevated ($p < 0.001$) in HIV-positive compared to HIV-negative individuals. We have added this point into the results section 'Relationship between HBV infection and liver disease' as follows: 'Individuals with HIV or HBV infection had higher liver function tests (ALT, AST, ALP, GGT) and elevated liver fibrosis scores (FIB-4, APRI, GPR, and S-Index) compared to uninfected individuals (Suppl Table 4).'

3. Do the authors find the GPR rate of advanced fibrosis of 25% (1:4) alarmingly high - is this perhaps an overestimation? Are there any other suggested reasons?

Yes, we agree this seems extremely high, and we have added a specific comment to highlight this (with reasons) in the discussion, as follows: 'The finding that almost 1:4 individuals in this population study has an abnormal GPR score is concerning and striking. This could be influenced by high GGT values (potentially in association with alcohol), or low platelet counts (for the reasons outlined above). We used stringent thresholds for GGT, with different thresholds for the upper limit of normal in males and females (Suppl Table 1), which will translate into a higher proportion of the population meeting the threshold for elevation of both GGT and GPR.'

Reviewer: 2 (Asgeir Johannessen, Oslo University Hospital)

This well-written study explored LFTs among a rural population in Uganda. The authors calculated various scores of liver fibrosis and tried to link this to demographic and other population characteristics. Since data is scarce on the performance of these liver fibrosis scores from rural Africa, the paper provides new and relevant information. The lack of other measures of liver fibrosis, however, is a clear limitation. Thank you for positive and balanced feedback.

Specific comments:

- Page 9, line 13-17: The most striking result is the very large variation in fibrosis scores, from 23.5% with GPR to 0.1% with RPR. I think this deserves a comment in the Discussion.

We have added to the first paragraph of the discussion to highlight this point specifically, as follows: 'Composite fibrosis scores have been developed with the aim of improving sensitivity of detection of liver disease (29), but these it is striking that there is a large variation in the prevalence of liver fibrosis estimated by different scores, ranging from 23.5% based on assessment using GPR, down to <1% with RPR.'

- Page 9, line 13-17: In the original paper by Lemoine et al, the GPR was based on a GGT threshold of 58 in both men and women, whereas you employ a much lower cutoff for women. This should be commented (in the Discussion) since it automatically results in a higher GPR result which can explain why you find a higher proportion above the threshold using this test.

We agree, and have added to the discussion to say 'We used stringent thresholds for GGT, with different thresholds for the upper limit of normal in males and females (Suppl Table 1), which will translate into a higher proportion of the population meeting the threshold for elevation of both GGT and GPR' (and this point has been incorporated in the modification we made in response to point 3 raised by reviewer 1).

• Page 9, line 24-28 (“Based on previous validation among African...”): This is very uncertain, since results are scarce and to a certain degree conflicting. See for example Desalegn et al, Liver Int 2017, where APRI, GPR and FIB-4 were quite similar in performance.

We have amended the sentence in question in the results section, such that we now clarify by saying there is ‘some limited evidence’ for the superiority of GPR as a fibrosis score. We have also improved the discussion to highlight this point: ‘However, the evidence is limited, and to some extent conflicting...’, and also adding the specific reference that has been suggested by the reviewer:

‘Meanwhile, a study in Ethiopia reported a similar specificity of APRI, GPR and FIB-4 for the detection of

fibrosis and cirrhosis [ref Desalegn doi: 10.1111/liv.13393].’

• Page 9, Alcohol paragraph: Don’t you have information about quantity of alcohol consumed? The categories you report are not very helpful in assessing whether these people have a harmful alcohol consumption, and thereby might have alcoholic hepatitis. You write in the discussion that the quantity of alcohol in local drinks is hard to estimate, but perhaps you have recorded how often they drink so that you can make a rough estimate? E.g. daily drinkers vs. the rest?

The data that we have regarding quantification of alcohol consumption are already presented in Suppl Fig 3, in which individuals are classified as drinking alcohol in 3 different categories (never / within the past 12 months / within the last 30 days). We agree that it would be desirable data to present more details, but we do not have any further information that we regard as being reliable pertaining to the specific volume / frequency of alcohol consumed. We have added a comment to the discussion as follows: ‘Further data collection to quantify the frequency, volume and nature of alcohol consumption will be important to improve insights into the relationship between alcohol and liver disease in our population setting.’

• Page 9-10: PAR: I think you overstretch your data here. With so blunt tools at hand, you should be very cautious about making a statement about PAR for any risk factor. I think the article is better if you skip it. You risk diluting your nice and valuable results with these very speculative (and probably erratic) PAR results.

We agree that these results must be represented with caution, and have reflected carefully on whether to remove the results of PAR analysis altogether. While we recognise the limitations of the approach, we also feel that the extent to which the PAR analysis highlights the impact of alcohol is important, as this is in keeping with the output from the rest of the paper and is an important finding for liver disease in this setting (and perhaps in others across sub-Saharan Africa), with the potential to inform interventions if replicated in other studies.

In response to the feedback received, we have made the following modifications:

• We have removed the sentence reporting PAR analysis from the abstract.

• We have added to the methods to describe the rationale / approach to PAR analysis, providing improved context to explain the approach, as follows: ‘We calculated population attributable risk (PAR) as the proportion of the cases of liver dysfunction (defined either as elevated LFTs or fibrosis score) in the population that is due to exposure to alcohol, HIV or HBV.’

• We have added a short paragraph to discussion to reflect the relevant considerations, as follows:

‘The calculation of PAR that we have undertaken

in this study should be interpreted with caution, as we recognise that robust assessment of exposure to alcohol is difficult, and the markers we are using to represent underlying liver disease each come with their caveats. We have nevertheless included this analysis as part of our output on the grounds that it is congruent with other aspects of the analysis in highlighting a likely significant role for alcohol as a driver of liver disease, and therefore may be of influence in informing future studies as well as underpinning appropriate interventions.’

We would be happy to rediscuss this point with the reviewers or editorial team if further amendment is thought to be required.

• Table 2: This table is hard to use for others who are interested in the subject. I suggest that you keep the same format, but instead of giving OR for abnormal test results (which is not very interesting) you can give the median result with interquartile range for each specific test. For example, for APRI you give the absolute APRI result for men and women, and then you can highlight whether the difference is statistically significant. By doing this, others can see the absolute APRI results and make up an idea of normal values in the different population groups.

We agree that it is certainly helpful to present absolute values, and we have added a new analysis to show these data (Suppl Table 4). This also supports the analysis of the impact of HIV status on fibrosis scores as suggested by Reviewer 1.

It should be noted that many studies rely on a binary classification of fibrosis scores above/below a specific threshold (rather than using these variables as a linear scale), so some caution is required when presenting fibrosis scores based on a continuous distribution. Odds Ratio relies on classifying fibrosis scores more conventionally as raised or not raised, and supports our presentation of key results in Figure 2, so we have also retained this component of the analysis. We would be happy to discuss this further if the editorial team or reviewers feel we have not struck the right balance in the choice of data to present.

• Page 10, “Abnormal LFTs...” paragraph: You repeat “compared to males” twice in the first sentence. Thank you for spotting this, we have removed the duplication.

• Page 10, “Relationship between BBV...” paragraph (and suppl fig 5): HIV infection in itself can give low platelets, and therefore all the fibrosis markers you investigate can be erratic in this group (see for example Johannessen A, Lemoine M. *Liver Int* 2015: 2059). And HIV treatment can give elevated LFTs, which means that these tests might be less accurate (or even useless?) in this population. Thus, your conclusion “Therefore, GPR and S-index may be the most sensitive markers of inflammation...” is not correct. I see that you address this in the Discussion, but you should avoid making this statement here.

We have removed the sentence ‘Therefore, GPR and S-index may be the most sensitive markers of inflammation in the context of HBV or HIV infection’.

• Page 10-11, Liver disease of unknown etiology chapter: Here you seem to assume that GPR (and the other NIVs) have 100% positive predictive value. But in a population-based cohort like yours, where the background prevalence of liver fibrosis is low, the majority of those with an abnormal test result will NOT have liver disease. You should acknowledge this also in the Results chapter.

We have added to this paragraph in the results section as follows: ‘We also recognise that in the setting of a population-based cohort (where the background prevalence of liver fibrosis is relatively low), many of those with an abnormal test result will not have liver disease; these false positive hits based on GPR may also account for the 66% in whom we could not identify a risk factor.’

• Page 13, 1st paragraph: You should also mention khat chewing (which is not a traditional medicine, but a very popular recreational drug), which was recently found to be a major cause of unexplained liver disease in east Ethiopia (ref: Orlien S, et al. *Hepatology* 2018: 248-257). And another study by the same group found that more than half of the liver disease in the area was not caused by alcohol or viral agents (Orlien S, et al. *BMC Gastroenterol* 2018;18:27).

Thank you; we have amended this paragraph in the discussion to incorporate these two references and to add specific reference to the use of Khat in some settings; the new text reads as follows: ‘This is in keeping with other studies from Africa that report a high proportion of cases of liver disease that are not attributable to viral infection or alcohol and could be as a result of other understudied factors such as NAFLD and use of traditional medicine (35)(Orlien S, et al. *BMC Gastroenterol* 2018;18:27). Khat chewing (a popular recreational drug in some settings), was recently found to be a major cause of unexplained liver disease in east Ethiopia (Orlien S, et al. *Hepatology*

2018: 248-257).’

- Discussion, last paragraph: In the last sentence you should change “liver fibrosis” to “liver fibrosis tests” since you provide only indirect evidence of liver fibrosis. We have amended as suggested, such that this now reads ‘...we have identified alcohol, HIV and HBV as risk factors for deranged LFTs and elevated liver fibrosis scores...’

Reviewer: 3 (CW Spearman, Faculty of Health Sciences, U of Cape Town)

This is a well written manuscript which provides further important insight into the need to assess the validity of fibrosis scores in Africa and the appropriate normal reference ranges for liver function tests in Africa.

Thank you for your positive feedback.

No specific amendments have been suggested by this reviewer.

VERSION 2 – REVIEW

REVIEWER	Mark Sonderup University of Cape Town, Medicine
REVIEW RETURNED	29-Jan-2020

GENERAL COMMENTS	Thank you for adequately addressing my comments/questions.
--

REVIEWER	Asgeir Johannessen Oslo University Hospital Ullevål Norway
REVIEW RETURNED	05-Feb-2020

GENERAL COMMENTS	The authors have responded to all my comments and the paper is now acceptable for publication.
--